# Double ridge formation over shallow water sills on Jupiter's moon Europa

Riley Culberg [1✉], Dustin M. Schroeder [1,2] & Gregor Steinbrügge[2]

Jupiter's moon Europa is a prime candidate for extraterrestrial habitability in our solar system. The surface landforms of its ice shell express the subsurface structure, dynamics, and exchange governing this potential. Double ridges are the most common surface feature on Europa and occur across every sector of the moon, but their formation is poorly understood, with current hypotheses providing competing and incomplete mechanisms for the development of their distinct morphology. Here we present the discovery and analysis of a double ridge in Northwest Greenland with the same gravity-scaled geometry as those found on Europa. Using surface elevation and radar sounding data, we show that this double ridge was formed by successive refreezing, pressurization, and fracture of a shallow water sill within the ice sheet. If the same process is responsible for Europa's double ridges, our results suggest that shallow liquid water is spatially and temporally ubiquitous across Europa's ice shell.

[1] Department of Electrical Engineering, Stanford University, Stanford, CA, USA. [2] Department of Geophysics, Stanford University, Stanford, CA, USA.
✉email: culberg@stanford.edu

Jupiter's icy moon Europa harbors a global subsurface ocean beneath an outer ice shell[1–3]. The thickness and thermophysical structure of this ice shell are poorly constrained, but models suggest it may be 20–30 km thick[4–8] with a layer of warm, convecting ice underlying a cold, rigid crust[9,10]. The detailed structure and dynamics of its ice shell and the timescales over which they evolve are critical for understanding both the fundamental geophysical processes and habitability of Europa[11]. Some of the primary observational constraints on these subsurface processes are their expressions in the surface morphologies imaged by Voyager and Galileo.

Europa's surface is young[12] and geologically active[13,14], displaying a wide variety of landforms including ridges, troughs, bands, lenticulae, and chaos terrain[15]. Of these, double ridges are the most common, consisting of quasi-symmetric ridge pairs flanking a medial trough[15,16], with height to peak-to-peak distance ratios <0.58[17]. These ridges may extend for hundreds of kilometers and include some of the oldest features visible on the surface, with frequent cross-cutting implying numerous formation cycles over Europa's history[15,16]. Cycloidal ridges and ridge complexes share many of these characteristics and along-strike transitions between ridge morphologies are not uncommon, suggesting that a single process may be active in the formation of all ridge types[18,19]. Proposed formation mechanisms for double ridges fall into six categories: cryovolcanism[20], tidal squeezing[13], diapirism[21], compression[22], dike intrusion or ice wedging[23,24], and shear heating[25,26]. All of these mechanisms require ice-shell fracture and, with the exception of compression and diapirism, all invoke near-surface ice-water interactions, either through internal melting[26] or direct injection from the subsurface ocean[13,20,24].

More recently, a number of extensions to the explosive cryovolcanism hypothesis have been proposed, in part to address the difficulty of driving negatively buoyant ocean water directly to the surface[27]. These models suggest that double ridges may instead form above shallow, crystallizing water bodies, such as sills or dikes, within the ice shell, rather than by direct connection to the subsurface ocean[19,28–30]. Such mechanisms are more consistent with morphometric analyses that both disfavor compressional formation models and support the presence of subsurface water reservoirs beneath ridges[28]. These models have much in common with formation mechanisms for lenticulae, chaos, and cratered terrain that invoke the emplacement and refreezing of shallow water bodies to explain the observed doming, surface disruption, or collapse[31–33]. This growing body of work suggests that shallow water may be critical, not only to double ridge formation, but also to Europan ice-shell dynamics, exchange, and ultimately habitability.

Analogs for double ridges or confined shallow water bodies from the terrestrial cryosphere could place powerful constraints on this hypothesis space for Europa[34]. Sea-ice pressure ridges[35] and some ice rise divides[36] bear a qualitative resemblance to Europa's double ridges, but the pressure ridge analog assumes a very thin ice shell and ice divides express a flow regime entirely dissimilar to the Europan environment. Similarly, subglacial volcanic craters and ice shelf brine infiltration have been invoked as analogs for pressurized ice-water interactions[32], but neither fully captures the physics of refreezing water bodies confined within an ice matrix.

Here, we present an icy double ridge discovered on the Greenland Ice Sheet with the same gravity-scaled geometry as Europa's double ridges (Fig. 1). High-resolution ice penetrating radar observations reveal that this ridge is underlain by a shallow refreezing water sill (Fig. 2) and provide a direct window into the subsurface processes that drove its formation.

## Results

**Greenland observations.** This double ridge is located ~60 km inland of the ice-sheet margin in Northwest Greenland (Supplementary Fig. 1) at 74.566° N, 54.0531° W. ArcticDEM digital elevation models (DEMs) (Supplementary Fig. 2) and ICESat-2 satellite laser altimetry measurements (Fig. 3d) show that an asymmetric precursor ridge system developed by July 2013, with the full double ridge forming by March 2016 and persisting to the present day. The most recent DEM, collected in March 2016, shows two ~800 m long quasi-symmetric ridges flanking a central trough. The average ridge height is 2.1 m, with the northern ridge ~1.4 times the height of the southern ridge. The peak-to-peak distance varies from 26 to 78 m along-strike, with an average width of 46 m. After scaling the mean ridge height by the ratio of Earth's gravity to Europa's, the average height to peak-to-peak distance ratio is 0.37, entirely consistent with Europan ridge geometry[17]. Surface imagery and the DEMs also show evidence of flexure along the ridge flanks (Fig. 1, Supplementary Fig. 2), similar to ridges observed on Europa[28].

The double ridge is located within the percolation zone where the ice-sheet near-surface consists of a layer of porous, compacting snow known as firn that reaches the density of solid ice some tens of meters below the surface. During the summer melt season, surface meltwater percolates into this porous layer, where it can refreeze to form multi-meter-thick, horizontally continuous ice slabs, extensive ice layers (0.1–1 m thick), or smaller ice lenses (<0.1 m thick) with limited horizontal extent. Previous work has shown that the subsurface structure in this area typically consists of a 3–5 m thick low-permeability ice slab perched above a partially porous firn layer with significant ice lensing that transitions to solid ice at a depth of 30–40 m[37,38]. At the local scale, the ridge is situated in the bottom of closed surface basin with an average ice velocity of 106 ma$^{-1}$, flowing approximately parallel to the strike of the ridge.

Ultrawideband airborne radar sounding data collected in May 2016 as part of NASA's Operation IceBridge using the Center for Remote Sensing of Ice Sheets (CReSIS) Multichannel Coherent Radar Depth Sounder (MCoRDS) reveal that this double ridge is underlain by a shallow water sill (Fig. 2). This cross-sectional subsurface image, oriented transverse to the ridge strike, displays anomalously bright subsurface echoes consistent with reflections from a water table located 10–15 m below the ice surface and extending laterally for ~1.6 km (Methods). This sill is well below the typical thickness of the overlying refrozen ice slabs[37], suggesting that it was originally emplaced within the relict porous firn layer. However, a localized surface uplift of up to 9 m associated with sill emplacement (Supplementary Fig. 3), as well as strong hyperbolic scattering from the water table (Supplementary Fig. 4), suggests that this water may occupy a macro-porous fracture or conduit network rather than simply saturating pre-existing pore space. Based on the displaced surface volume, we estimate the initial water volume was at least $7.2 \times 10^6 \pm 1.7 \times 10^6$ m³ (Methods). The sill is surrounded by an ellipsoidal region of extremely low relative radar reflectivity and minimal stratigraphic structure, consistent with a homogeneous region with minimal internal contrast in material density (Fig. 2). This suggests that the sill is encased within a refrozen ice mass that extends to a total width of ~2.4 km and a depth of about 25 m below the surface. Pore close-off is estimated to occur at ~30 m depth in this region[38], so this sill likely rests on an impermeable base of meteoric ice. We also observe a distinct upwelling of the water table directly beneath each of the ridges, with particularly low radar reflectivity beneath the central trough, suggesting a break in the sill.

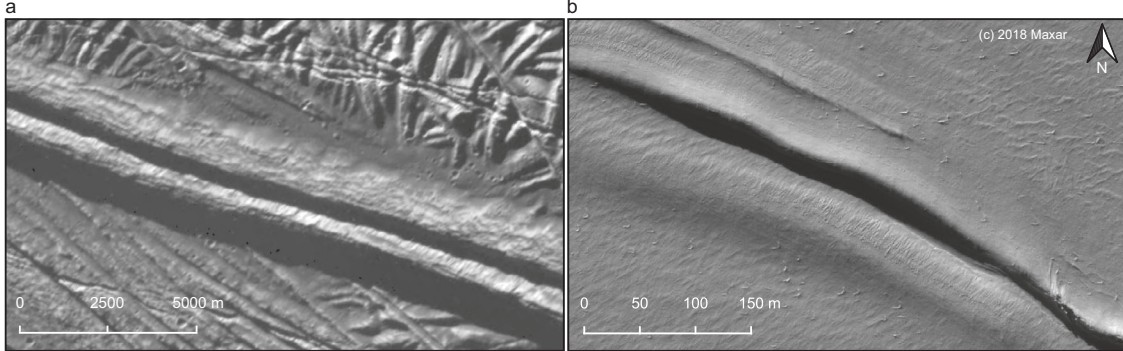

**Fig. 1 Surface imagery comparison of a double ridge on Europa and Earth. a** Europan double ridge in a panchromatic image from the Galileo mission (image PIA00589). The ground sample distance is 20 m/pixel. **b** Greenland double ridge in an orthorectified panchromatic image from the WorldView-3 satellite taken in July 2018 (© 2018, Maxar). The ground sample distance is ~0.31 m/pixel. Signatures of flexure are visible along the ridge flanks, consistent with previous models for double ridges underlain by shallow sills[28].

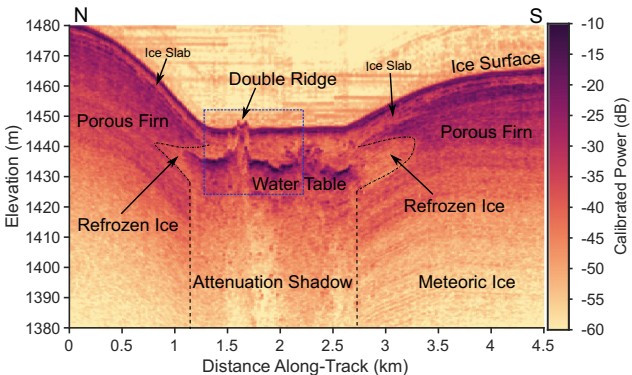

**Fig. 2 May 2016 MCoRDS radargram showing the double ridge.** The color scale shows the calibrated received radar power with darker colors indicating a stronger return. The dashed blue box outlines the region shown in more detail in Fig. 3. This radargram shows the full crystallizing sill perched within the porous firn beneath the ice slab and resting on a base of impermeable meteoric ice. The water table is marked by a strong, horizontal, subsurface reflection below which no structure is visible due to the high attenuation. A clear split in the water table is visible directly beneath the double ridge. The surrounding region of low reflectivity is the refrozen ice shell.

To constrain the degree to which each region has been modified by surface meltwater infiltration and refreezing, we use a Markov Chain Monte Carlo inversion with a radar scattering forward model to map the distributions of radar reflectivity within the firn, ice shell, and sub-ridge regions to estimates of the fractional refrozen ice volume in each area (Supplementary Fig. 5, Methods). This also provides a proxy for porosity, with higher ice content reflecting lower porosity. Consistent with our interpretation that the sill is encased in a refrozen ice shell, we find an ice fraction of 0.8 (0.27) (median and interquartile range) for the low reflectivity regions surrounding the sill, compared to 0.65 (0.28) in regions identified as firn. We interpret an additional 5 dB reduction in radar reflectivity directly beneath the ridge system, corresponding to a median ice fraction of 0.86 (0.23), to show that the sill was split by refreezing within a vertical conduit or fracture network, concentrating subsequent water injection from the sill into the firn directly to either side of this now impermeable barrier.

We directly observe these subsurface processes associated with ridge formation in a series of high-resolution radargrams collected by the CReSIS Snow Radar between May 2015 and March 2017 (Fig. 3). In the May 2015 data (Fig. 3a), we observe a

highly asymmetric system with a prominent northern ridge ~2 m in height and minimal evidence for a southern ridge. The water table is relatively continuous, peaking to the north of the ridge, and the sub-ridge upwellings have not yet developed. The May 2016 data (Fig. 3b) shows a clear double ridge with the divided sill and symmetric upwellings directly beneath each ridge. This radargram also shows a new set of stacked point scatterers beneath the central trough and capping the divide in the sill (Supplementary Fig. 4). These signatures are consistent with scattering from some form of englacial void space (Methods), supporting our interpretation that the sill was divided after water refroze within a central vertical conduit or fracture network. The March 2017 data (Fig. 3d) reflects a similar morphology, although we observe small variations in the ridge height and a reduction in the relative reflectivity of the upwelling portion of the water table as well as the central point scatterers that could be consistent with ongoing refreezing or creep closure of voids. However, given that the ridge is advected past the fixed radar flight path at a rate of ~106 ma$^{-1}$ and the spatial variations in ridge height and width observed in the 2016 surface DEM, these changes in observed morphology and reflectivity may reflect along-strike variability in ridge structure, rather than a true change in ridge structure with time.

**Ridge formation mechanism.** These observations reveal a refreezing-driven mechanism behind the formation of this double ridge (Fig. 4). In the first stage, a shallow water sill is emplaced within a porous matrix perched on a much thicker base of impermeable ice (Fig. 4a). In Greenland, seasonal melting, surface topography, and stress regime likely played critical roles in sill emplacement. We infer that during the 2012 extreme melt season, meltwater flowed laterally over the ice slab surface into the surface basin hydraulic low before ultimately draining into the subsurface through fractures in the overlying ice slab to form the sill observed in the radar data (Methods). Leak-off into the porous firn layer may have reduced the water pressure within the fracture enough to prevent fracture propagation to the ice-sheet bed as is common in similar closed basins in Greenland's ablation zone. However, mechanical failure of the overlying firn or ice slab during injection and uplift may have led to the formation of the initial fracture observed near the edge of the uplifted region (Supplementary Fig. 2b).

In the second stage, inflow or melting ceases and the sill begins to refreeze from its outer boundaries inward, eventually leading to upward-propagating fractures in response to overpressure in the confined water body[31,39] (Fig. 4b). In Greenland, overpressure may have reactivated the initial fracture observed in 2013. Water

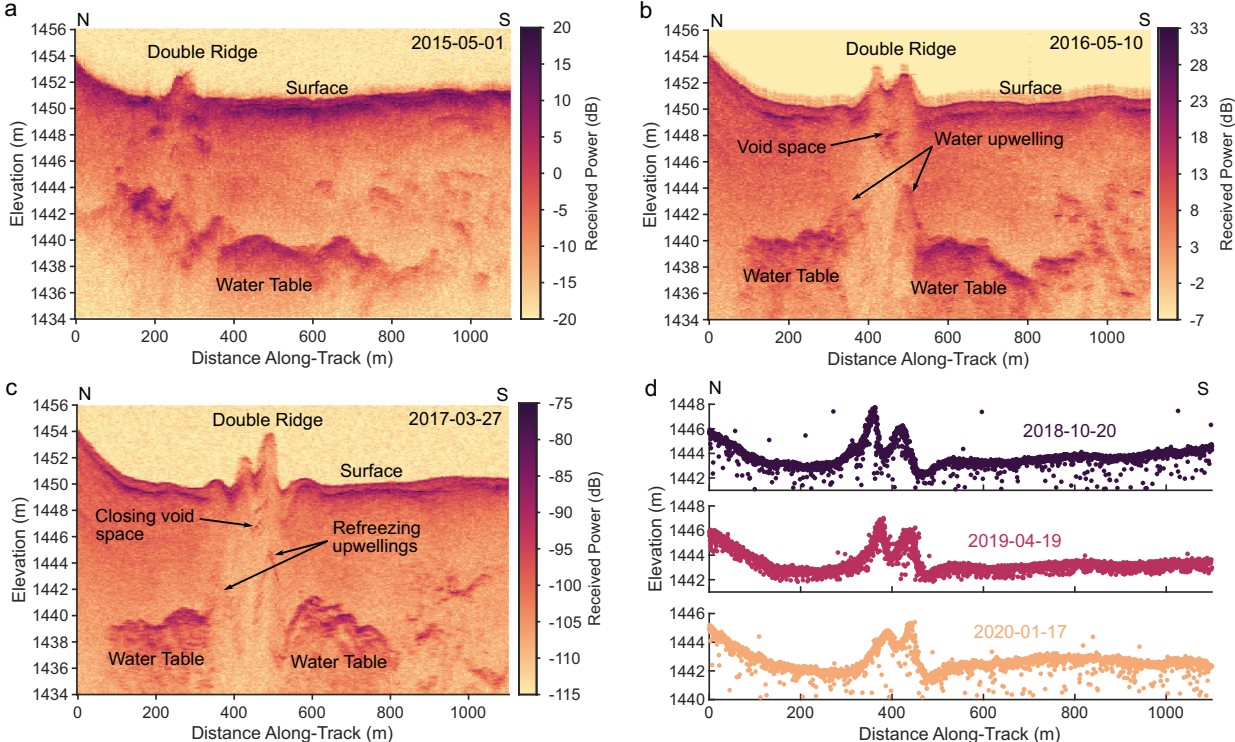

**Fig. 3 Time series of high-resolution radargrams and ICESat-2 laser altimetry showing the evolution of the double ridge system.** In panels **a–c**, the color scale shows the relative received radar power with darker colors indicating stronger returns. In panel **d**, the surface elevation profiles are color-coded by collection date. **a** Radargram collected in May 2015. The water table is continuous and the southern ridge has not yet developed. **b** Radargram collected in May 2016. This radargram shows the full double ridge, underlain by the divided and upwelling water table, along with scattering from likely void space beneath the central trough. **c** Radargram collected in March 2017. The system morphology remains generally stable, with reduced reflectivity in the upwelling water table and void spacing, suggesting freeze-out and creep closure. **d** ICESat-2 surface elevation profiles collected over the ridge in October 2018 (dark purple), April 2019 (light purple), and January 2020 (orange). These plots show high-quality (level 4) photons from ground track 331. Apparent changes in ridge cross-section may be the result of along-strike variations in ridge structure as the ridge is advected ~106 ma$^{-1}$ under the fixed satellite track.

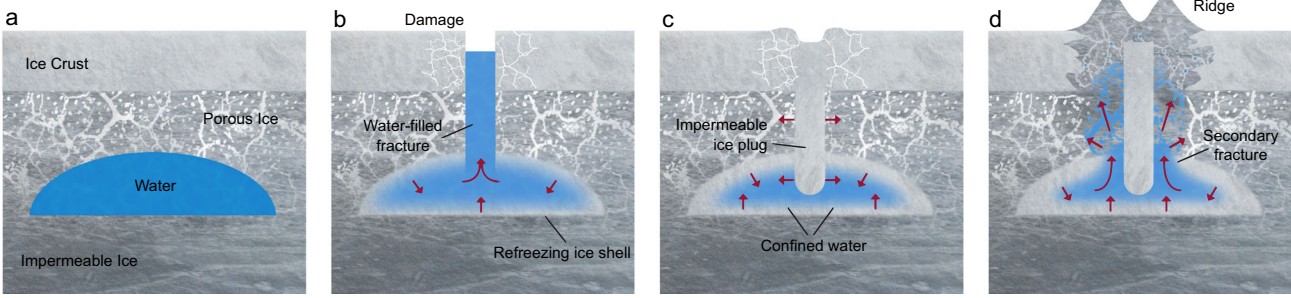

**Fig. 4 Double ridge formation mechanism (not to scale).** Gray areas are ice, white/light gray shows pore space, and blue areas are liquid water. Red arrows show the direction of forces acting on the sill and the direction of water flow if relevant. **a** Stage 1—sill emplacement. In Greenland, this likely occurs through concentrated drainage of surface meltwater[41]. On Europa, sills might form by direct injection from the subsurface ocean[27,29,43], from melting over rising diapirs[32,44], or through shear heating[26,30]. **b** Stage 2—the sill fractures due to internal overpressure or other near-surface stresses and water fills the central conduit. **c** Stage 3—the sill is divided by refreezing within the central conduit or fracture network. **d** Stage 4—a double ridge develops as overpressurized water within the refreezing sill is forced out along the planes of weakness flanking the refrozen central conduit, producing the symmetric surface doming.

then rises within these fractures due to the pressure gradient and, in stage three, refreezes, forming an impermeable ice plug that divides the sill in two (Fig. 4c). In Greenland, although this initial fracture formed along the edge of the sill, the system still equilibrated to the classic quasi-symmetric double ridge morphology, except where the fracture was complex or discontinuous. This suggests that ridge symmetry may be a result of the internal physics of formation rather than the initial geometry of the

system, and that this mechanism is robust to fracture position and orientation so long as some minimum water volume is retained within both fragments of the divided sill. Similarly, if some hydrologic connection is maintained between the two sill segments, either because the central refrozen conduit does not extend the full depth of the sill or because it is discontinuous along-strike, this would ensure that the pressure gradient is similar on both sides of the central refrozen conduit, leading to

similar degrees of surface extrusion. While we also observe a transition from single ridge to double ridge in Greenland during these stages, it is not clear if this evolution is an innate feature of the mechanism we describe here, or specific to some local condition such as an initial asymmetry in pressure between the two sill segments or the external influence of transverse compression in this surface basin. The single ridge is unlikely to be an expression of the central refrozen conduit itself, given that there is no clear mechanism for such a ridge to then subside to form the medial trough between the double ridges. However, our observations do suggest that it is possible for single ridges to later evolve into double ridges under these general conditions.

In the final stage, ongoing refreezing continues to raise the pressure in the sill, producing secondary fractures along the existing planes of weakness at the rheologic boundaries to either side of the refrozen central conduit (Fig. 4d). This creates high permeability pathways into the overlying porous matrix, and water preferentially flows along these paths in response to the pressure gradient, creating the observed upwellings and injecting water into the overlying material. A combination of volume expansion as this water refreezes, the subsurface pressure gradients within the sill itself, and lateral compression from refreezing in the central conduit[19] ultimately force the softer, porous material up and out along the weak points to either side of the refrozen central conduit, extruding at the surface to form the double ridge. In this model, volume expansion from refreezing likely generates the added volume of the ridges. In Greenland, approximately $18.6\%0ex+5.7\%-3.5\%$ of the initial sill volume would need to refreeze to form the ridges as observed in 2016, consistent with the large refrozen ice shell we observe around the liquid water table. Widespread surface subsidence over the sill or sill recharge during the melt season could also allow for some of the implied redistribution of mass or addition of volume. However, the registration accuracy of the available DEMs and poorly constrained accumulation, ablation, and ice flow processes preclude any reliable partitioning of surface elevation changes, which would be needed to assess whether these processes occurred in Greenland.

**Greenland as a Europa analog**. Our results suggest that the refreezing of confined aquifers in the near-surface of the Greenland Ice Sheet is a promising analog for the progressive pressurization and fracture of crystallizing shallow sills in the brittle lid of Europa's ice shell. In particular, Greenland provides examples of the evolution of isolated liquid water pockets confined within a semi-permeable, subfreezing ice matrix and perched on an impermeable ice body of much greater thickness, similar to hypothesized conditions at Europa. However, significant differences in gravity, atmospheric pressure, temperature, and ice impurity content will likely lead to complex scaling relationships between the two environments. While the dearth of observations at Europa limits our capacity to constrain many of these variables, in the context of double ridge formation, we can assess the general viability of the physical processes we invoke and the qualitative impact of these environmental differences.

In general, the low gravity (1.3 m/s) and atmospheric pressure (0.1 µPa) at Europa should aid ridge formation. Both conditions act to reduce overburden on the sill, relaxing the overpressure conditions needed to drive water towards the surface or uplift the material overlying the sill. A smaller fraction of the sill might be required to refreeze to form the ridges in this case, but only if the degree of refreezing needed to achieve the minimum overpressure exceeded the degree of refreezing needed to generate the added ridge volume by expansion.

Ice temperature and chemistry are poorly constrained confounding factors at Europa that play an important role in governing ice rheology. Our model invokes some degree of near-surface porosity, suggesting that sills would need to form within or at the lower boundary of the porous portion of Europa's ice shell. Previous work estimates this near-surface layer to be on average 3 km thick[8,40], and most models of sill emplacement on Europa have assumed depths 1–5 km for sill intrusions[19,28,29,31]. This would place sills firmly within the brittle, conducting portion of the ice shell where temperatures likely do not exceed 150 K (ref. [8]), except where latent heating can warm the surrounding ice. Therefore, we expect elastic deformation or brittle fracture to dominate in response to imposed stress, with little to no contribution from viscous creep. This is consistent with our proposed mechanism, which invokes fracture and rapid uplift or extrusion of damaged material to form the ridges, despite the significantly warmer conditions in Greenland (~260 K). However, the length scales and timescales for ridge formation would be strongly modulated by the details of the rheology.

These temperature differences also imply an absence of surface melting on Europa, whereas melt is common at the double ridge site in Greenland. In particular, Landsat 7 ETM+ and Landsat 8 OLI optical satellite imagery show visible surface runoff and localized water ponding within the basin during four out of the five summers spanned by our DEM and radar observations (Supplementary Fig. 6). Therefore, we consider two key ways in which the frequent presence of surface water might limit the value of the Greenland double ridge system as a Europa analog.

First, seasonal runoff might recharge the sill, either by activation of old fractures or formation of new fractures during the melt season. Sill replenishment would lower the total amount of refreezing required to form the new ridge volume and lengthen the refreezing time relative to a closed system, given the seasonal influx of additional mass and heat. However, this would not necessarily prevent the cycle of refreezing, fracture, and pressurization proposed for ridge formation, and in fact, sill replenishment from the subsurface ocean has also been considered as a viable mechanism for prolonging sill lifetime on Europa[29].

Second, surface melt, stream incision, or preferential ablation beneath ponded water might actively rework Greenland's surface topography, leading to ridge morphologies that would not be possible on Europa. However, we assess that it would be difficult to form isolated ridges by preferential downcutting of the surrounding terrain unless the ridges were already of a sufficient height to prevent ponding or flow overtop of them. Additionally, within a closed basin, heterogeneous melting should primarily redistribute topography laterally, making widespread surface lowering by downcutting difficult to achieve. We also observe a largely stable ridge morphology in the DEM, radar, laser altimeter, and surface imagery observations between 2016 and 2020, despite significant differences in annual accumulation, melt, and surface runoff from year to year (Supplementary Fig. 6). This suggests that the double ridges are unlikely to be predominantly shaped by ongoing seasonal melt or water flow. In general, it is more likely that surface processes would reduce, rather than heighten, the symmetry and along-strike consistency of the Greenland ridges, reducing rather an increasing the similarities with Europa's ridges. The absence of these processes on Europa might explain the more consistent dimensions of the moon's ridges.

Thus, despite the differences in environment, we suggest that Greenland can offer promising analogs for shallow water dynamics in the brittle lid of Europa's ice shell. The Northwest Greenland marginal accumulation zone is of particular interest due to its abundance of near-surface refreezing and liquid water features, including ice slabs[37], firn aquifers[41], and buried supraglacial lakes[42]. This environment is a natural laboratory

for studying the surface expressions and underlying dynamics of near-surface water reservoirs confined within a shallow, sub-freezing porous ice matrix that transitions to impermeable ice with depth. Further investigations into the effects of water migration, refreezing, and fracture on the permeability and rheology of the subsurface, as well as the interplay between surface stresses and subsurface pressure gradients, may offer new insights into the dynamics of other Europan features such as lenticulae and chaos terrain.

## Discussion

This work provides the first direct observations of double ridge formation over a crystallizing shallow water body and the inter-actions between surface fracture and refreezing that drive the characteristic morphology. If this mechanism controls double ridge formation at Europa, it suggests that the process does not require direct ocean-to-surface communication via explosive cryovolcanism[20] or tidal squeezing[13]. However, injection from an overpressurized ocean[27,29,43], shear heating[26,30], or melt over rising diapirs[32,44] might produce the shallow water sills we invoke, implying that some form of dynamic transport within the ice shell is still necessary and common. Additionally, cryomagma extrusion at the surface would be possible if the overlying matrix is of sufficiently low strength, which may explain the variable presence and absence of surficial deposits on the flanks of Europa's double ridges[16]. In general, our observations of double ridge formation in Greenland are most compatible with the physical processes of the Manga & Michaut model for lenticulae formation[31,43], which invokes the progressive refreezing and pressure-induced fracture of sills to explain surface deformation and disruption. This would place double ridges along a con-tinuum of surface features whose morphology may be a direct expression of the depth, volume, geometry, and refreezing rate of shallow water reservoirs, states which are in turn coupled to the rheology, physical structure, stress regime, and thermal state of the upper few kilometers of the ice shell.

For example, our proposed mechanism requires the develop-ment of both sills and linear surface fractures of at least the same length as the final double ridge in one dimension, which may be several hundreds of kilometers on Europa[15]. One possibility is that ridges form where surface fractures intersect circular sills whose diameters are on the order of the ridge length. However, given that chaos terrain may reflect complete surface collapse over shallow water bodies of only ~80 km in diameter[32], it might be difficult to plausibly support ridges hundreds of kilometers in length by this means without other surface disruption. On the other hand, if sills form by lateral spreading of ocean water that rises through fractures in the ice shell[27,29,43], elongated sills might form if continuous linear fractures of the same length as the ridges themselves can routinely penetrate through most of the ice-shell thickness. Alternately, elongated sills might be expres-sions of melt due to linear diapirism[16] coupled with a consistent or slowly varying depth to the eutectic point. If the central sill-splitting fracture forms due to internal overpressure, then it will likely follow the initial sill geometry as well. Alternately, tidal-stress-induced fractures might intersect a pre-existing sill[45] in more complex ways. In all cases, our mechanism relies on the routine formation of continuous linear fractures within the ice shell, implying significant lateral homogeneity in subsurface ice-shell rheology and composition would be necessary for Europa's ridges to form in this way. This stands in contrast to the strong local controls, including water availability and surface topo-graphy, that limit the extent of the Greenland double ridge to <1 km.

Similarly, the two orders of magnitude difference in ridge height and width between Europa and Greenland may be an expression of different limits on sill volume and depth. Under the proposed formation mechanism, ridges are built from material displaced by the expansion of the refreezing sill, suggesting that ridge size is modulated in part by the total amount of refreezing and the volume of near-surface material available to be displaced. Thus, Europa's larger ridges are likely supported by larger and deeper sills. While a full thermomechanical model would be required to establish a robust scaling relationship, a simple extrapolation of the ratio of the Greenland ridge's cross-sectional area per unit length to sill cross-sectional area per unit length suggests that Europa's ridges might be supported by water bodies on the order of $10^6 \, m^2$. This 2D volume estimate is highly con-sistent with predictions from analytical models of sills formed by injection from the subsurface ocean[43], and for a 500 km long ridge implies a total volume of 500 $km^3$, still two orders of magnitude smaller than the melt lens volumes previously esti-mated for Europa's chaos terrain[32].

Finally, our model explicitly requires some degree of near-surface porosity within the ice sheet or ice shell, which is expected in Greenland due to the ongoing accumulation and compaction of snow, but less certain on Europa. However, given the large degree of meltwater infiltration and refreezing in this region of Greenland, we estimate a porosity of only ~15% (based on the fractional ice volume estimates presented in Supplementary Fig. 5), which is within the range that has been previously pro-posed for the upper few kilometers of Europa's ice shell[32,40,46,47]. Interestingly, the numerous cross-cutting relationships between Europan double ridges would imply some mechanism for the eventual re-establishment of near-surface porosity after the initial pore space is filled by refreezing, suggesting that the ongoing development of a regolith[40,46] could be an important factor in ridge formation on Europa.

Altogether, our observations provide a mechanism for sub-surface water control of double ridge formation that is broadly consistent with the current understanding of Europa's ice-shell dynamics and double ridge morphology. If this mechanism controls double ridge formation at Europa, the ubiquity of double ridges on the surface implies that liquid water is and has been a pervasive feature within the brittle lid of the ice shell, suggesting that shallow water processes may be even more dominant in shaping Europa's dynamics, surface morphology, and habitability than previously thought.

## Methods

**Surface morphology.** We use strip-map ArcticDEM[48] digital elevation models (DEMs) to characterize the evolving surface morphology of the double ridge between 15 April 2012 and 25 March 2016. The strip-map ArcticDEM products are stereo-photogrammetric DEMs with 2 m posting produced from satellite image pairs collected by the WorldView-1, WorldView-2, and WorldView-3 satellites. We select DEMs with quality flags that exceed 0.98 (out of 1). The DEMs are provided with precalculated offsets to the x, y, and z spatial coordinates determined by co-registration to ICESat laser altimetry tracks, which we apply to the data to reduce absolute positioning errors in all three dimensions. The median relative vertical uncertainty is estimated to be on the order of 20 cm for these products after co-registration[49]. We measure the ridge surface morphology (ridge heights and peak-to-peak distances) from the 25 March 2016 DEM.

Coulter, et al. (2009) demonstrated that Europa's ridges have a consistent relationship between ridge height and width, characterized by the ratio of the mean ridge height to peak-to-peak distance[17]. To compare the geometry of the Greenland ridge to those on Europa, we calculate this same ratio for the terrestrial double ridge. The measured ridge height is 2.1 ± 0.57 m, with the northern ridge ~1.4 times the height of the southern ridge. The peak-to-peak distance varies from 26 to 78 m along-strike, with an average width of 46 m. We scale the measured height of the ridge by the ratio of the gravitational acceleration on Earth to that on Europa (ratio of 7.45), as any ridge-building mechanism would require uplift in direct opposition to this force. After scaling, we find a height to peak-to-peak ratio of 0.37 ± 0.17 ($\mu \pm \sigma$), which falls comfortably within the range of values calculated for Europa's ridges[17].

**Radar sounding data**. We use radar sounding data from two different systems flown as part of NASA's Operation IceBridge and built by the University of Kansas Center for the Remote Sensing of Ice Sheets (CReSIS). Deep sounding data were collected on 10 May 2016 using the Multichannel Coherent Radar Depth Sounder (MCoRDS)[50] operating at a center frequency of 300 MHz with 300 MHz of bandwidth, giving a range resolution of 0.42 m in solid ice and 0.51 m in firn (assuming a firn refractive index of 1.5). These data were focused using F-K migration with a 2.5 m target azimuth resolution. As displayed in Fig. 2, these data have been incoherently averaged 11 times with a moving window and decimated by a factor of 6 for ~30 m along-track resolution and ~15 m trace spacing. When analyzing the radar reflectivity, we do not apply additional incoherent averaging after focusing and retain the native 2.5 m trace spacing of the focused data.

The time series radargrams in Fig. 3 were collected by the CReSIS Snow Radar[51]. In 2015 and 2016 this system operated at a center frequency of 5 GHz with 6 GHz of bandwidth for a vertical resolution of ~2.2 cm and ~3.8 cm in ice and firn, respectively. These data were coherently averaged 32 times and incoherently averaged 11 times with a moving window before decimation by a factor of 5 to give ~10 m along-track resolution and ~5 m trace spacing. In 2017, the system operated with a center frequency of 10 GHz and 16 GHz of bandwidth for a vertical resolution of ~0.9 cm and ~1.4 cm in ice and firn respectively. These data were coherently averaged 32 times and incoherently averaged 11 times with a moving window before decimation by a factor of 5 to give ~10 m along-track resolution and ~5 m trace spacing.

With the exception of the 2015 Snow Radar data, all radargrams were elevation corrected using the radar-derived flight clearance and aircraft altitude from the system GPS and assuming an index of refraction of 1.78 (solid ice) in the subsurface. Due to significant along-track positioning errors in the 2015 Snow Radar GPS data, we use the 2015 ArcticDEM surface elevation extracted along the flight line for elevation correction in this season.

**Radar calibration**. We absolutely calibrate the 2016 MCoRDS radar data by correcting the received radar power for geometric spreading loss, system offsets, and englacial attenuation. We geometrically correct the data using a range squared dependence[52]. To find the system calibration constant, we cross-level this flight line with two interior tie-lines that intersect the sites of the B29[53] and B16[54] firn cores of the North Greenland Traverse. We then absolutely calibrate the data against simulated subsurface reflectivities using high-resolution density measurements from these cores[55]. We correct for englacial attenuation using the average firn conductivity from the B26–B29 firn cores[56–59] and an englacial temperature profile[60] simulated by the regional climate model MARv3.5.2[61]. Details of these correction and calibration procedures and associated sensitivity tests are described in the Supplementary Methods. We do not calibrate the Snow Radar data as the standard methods for absolute calibration assume that surface roughness is a small fraction of the radar wavelength[55,62], which is unlikely to hold for radars with centimeter-scale wavelengths.

**Water table interpretation**. The presence of subsurface liquid water is the only reasonable interpretation for the observed reflectivity and attenuation signals of the interface we identify as a water table. This interface has a calibrated reflectivity of −12.1 (7.468) dB (median and interquartile range), with 62% of the observed values falling between the theoretical reflectivity of a specular interface between ice and saturated firn (−14 dB) and the theoretical reflectivity of a specular ice-water interface (−3.5 dB)[63]. This distribution of observed reflectivities may already be an underestimate of the true permittivity contrast, as we observe significant off-nadir returns that suggests that some power was lost to radar scattering at a rough ice-water interface, although our synthetic aperture processing partially mitigates this effect[63]. The sudden disappearance of otherwise conformable englacial stratigraphy directly below this reflector in the 2017 MCoRDS data suggests significant attenuation of the radar waves as well. To estimate attenuation, we select five englacial reflectors located 100–200 m below the water table and compare their average power over 1 km segments to the left and right of the water table with the average received power directly beneath the water table at the equivalent depth. We observe an average power reduction of ~11 dB beneath the water table, relative to the surrounding ice. Assuming that the bottom of the refrozen ice shell extends no deeper than observed bottom of the refrozen ice shell, this implies an additional 11 dB of attenuation occurs over 15 m depth, equivalent to an englacial attenuation rate of 367 dB/km. Liquid water is the only reasonable mechanism for this absorption loss, which is almost 22 times greater than the typical depth-averaged englacial attenuation rates of 15–20 dB/km in this region[64].

The only other physical configuration, which could plausibly produce the observed mean reflectivity, would be if the subsurface contained a massive air-filled void, rather than a water body. However, it is physically unlikely that a $7.2 \times 10^6$ m$^3$ void could be formed and maintained for years without surface collapse, and ~41% of the reflectivity observations actually exceed the theoretical reflectivity of a specular air-ice interface. Additionally, void space could not produce the observed attenuation beneath this feature.

**Surface elevation change and ridge volume**. Assuming that the displaced surface volume is equal to the injected subsurface water volume for sill-like features[65,66],

we estimate the initial sill water volume based on the surface displacement. We measure the surface displacement associated with sill emplacement by differencing the co-registered DEMs collected on 15 April 2012 and 9 July 2013. We estimate the uncertainty in absolute elevation change by adding in quadrature the 70th percentile of the absolute vertical deviation from the ICESat registration track (approximately equivalent to the first standard deviation of the residuals) for each DEM as provided with the product registration files. The absolute uncertainty in surface elevation change for this DEM pair is estimated to be ±1.18 m. We manually select the bounding outline where the minimum surface displacement occurred and discretely integrate the surface elevation change within this boundary to estimate the displaced surface volume. We estimate uncertainty by calculating the total displaced volume assuming the surface displacement was either 1.18 m greater or less than measured. This volume estimate is likely an underestimate as it does not account for leak-off into the surrounding firn pore space. This method results in an estimated initial sill volume of $7.2 \times 10^6 \pm 7 \times 10^6$ m$^3$.

We assess that subsurface water injection is the most likely source of the observed uplift as we observe a subsurface water reservoir and the uplift occurred during or immediately after the extreme melt season in summer 2012[67]. The observed maximum uplift of 9 m is far in excess of the annual accumulation simulated by MARv3.5.2 in this region, which is on the order of 1.2 m assuming a snow density of 0.3 gcm$^{-3}$ and no compaction or melt. Ice flow also is unlikely to produce this kind of singular, localized uplift over a horizontal length scale less than the mean ice thickness, particularly as surface basins are understood to be expressions of stable subsurface topography.

We estimate the total volume of the ridge from the 2016 DEM. We extract the portion of the DEM within 500 m of the outer edges of the ridge system, excluding the region occupied by the ridge. We fit a mean plane to this surface, which we subtract from the full DEM to produce a map of local elevation relative to the mean surface of the basin. We then integrate the total volume within the area manually delineated as containing the double ridge to determine a total volume of $1.2 \times 10^5$ m$^3$.

**Reflectivity inversion**. To test our hypothesis that the sill is encased in a refrozen ice shell, with concentrated water injection and refreezing directly beneath the ridges and medial trough, we invert the observed radar reflectivity in each region for fractional ice content using an electromagnetic forward model that parametrizes radar reflectivity in terms of the density and geometry of the subsurface[55]. Based on the horizontal radar coherence and field observations from Greenland's ice slab areas[68,69] (see Supplementary Methods), we model the radar-illuminated subsurface footprint as a volume of solid ice containing discrete patches of relict layered firn. The effective reflectivity of the footprint can then be approximated as the coherent, area-weighted sum of the reflectivity of each firn or ice patch within the volume[52,63,70], neglecting interactions between patches. We parameterize the structure of each firn patch with mean firn density, mean firn density variability, and percentage of the total thickness occupied by ice layers, hereinafter referred to as the melt feature percentage (MFP). A single realization of footprint effective reflectivity is then calculated by randomly choosing the number and area of firn patches such that they cover the desired fractional firn area, generating a random vertical density profile for each firn patch based on the model parameters, simulating the effective reflectivity of each firn patch with a 1D layered dielectric reflection model[71], and coherently summing the reflectivity of all patches. The Supplementary Methods discusses the implementation of this model in further detail.

We establish distributions of radar reflectivity associated with the firn, refrozen ice shell, and sub-ridge material using the SAR-focused 2016 MCoRDS radargram. Supplementary Figure 5a shows the regions selected as representative of firn, refrozen ice shell, and sub-ridge material, and Supplementary Fig. 5b shows the associated distributions of reflectivity for each material type. We invert the mean reflectivity of each region, using the standard deviation of each reflectivity distribution as the uncertainty.

As our forward model is both non-linear and stochastic, we use the Markov Chain Monte Carlo (MCMC) method with a Metropolis-Hastings sampling scheme for inversion. This allows us to estimate full empirical distributions of model parameters that are consistent with the observed radar reflectivity distributions. Supplementary Table 1 lists the model and inversion parameters and their values. We assume uniform priors on the four model parameters and use a Gaussian likelihood function for proposed model evaluation. For each inversion, we run 1,000,000 iterations and remove the first 3000 as burn-in. We then calculate the fractional ice content, $F_{ice}$, from the estimated model parameters following equation (1), where $V_{radar}$ is the volume of the illuminated radar footprint, $A_{firn}$ is the fractional surface area of that footprint containing layered firn, and MFP is the fractional height of ice layers within those firn patches.

$$F_{ice} = \frac{[V_{radar} - A_{firn}(1 - MFP)V_{radar}]}{V_{radar}} \quad (1)$$

We assume that the focused footprint can be approximated as a rectangle whose width is set by the azimuth resolution, length is set by the diameter of the first Fresnel zone, and height is set by the radar range resolution. We also conduct sensitivity tests to determine the impact of changes in our assumptions about the

illuminated volume, surface roughness, or background firn parameters on the inversion results. These tests are discussed in the Supplementary Methods.

**Fracture analysis**. We reprocess the 2016 Snow Radar data without any along-track averaging (either coherent, or incoherent) in order to emphasize hyperbolic scattering from point features and rough surfaces. This data, with an along-track trace spacing of ~0.15 m, is shown in Supplementary Fig. 4. Directly beneath the central trough we observe ~8 stacked hyperbolic returns over a depth of ~2 m, with small lateral offsets over a total horizontal distance of ~32 m. Despite the incoherent scattering, these returns are still around 10 dB brighter than surrounding material, and of similar absolute power levels as the surface and upwelling water table, suggesting a significant dielectric contrast at the point of scattering. The hyperbolic returns are indicative of geometries with wide angular scattering functions such as spherical volumes, rough surfaces, or irregular volumetric geometries such as fractures and void space. Together this, suggests that the scattering comes from either air or water-filled pockets within solid ice.

The presence of liquid water in late spring at these shallow depths is less likely than void space due to thermal constraints. The radar data we analyze were collected in May 2016, just prior to the onset of the melt season, so any liquid water would have needed to survive the preceding winter without refreezing. The observed scatterers are located ~2–4 m below the surface, and firn temperature profiles from ice slab regions in Southwest Greenland[68] suggest that this is a sufficiently shallow depth that these features would be subject to seasonal temperature variations in response to changing atmospheric temperatures. For liquid water to persist at these shallow depths over winter would therefore require a sufficiently large water volume that latent heat exchange could prevent complete refreezing. However, the thermal profile of this area is likely complicated by seasonal surface ponding and ongoing refreezing of a large subsurface water body, which might result in warmer firn at shallower depths and make shallow water storage more viable than typically assumed. On the other hand, both surface and snow bridged crevasses commonly produce similar signatures in ice penetrating radar data, either due to diffraction where the fracture plane meets the surface, or scattering from irregularly shaped snow caps over subsurface void space[72]. Therefore, we interpret these returns as most consistent with scattering from some form of void space beneath the central trough. This could include scattering from different corners of an irregularly shaped void over a single conduit that was not fully filled by pressurized water, or diffraction at the tips of multiple adjacent, narrower fractures. Regardless, even in the case that liquid water is present, the distinct and separable nature of the hyperbolic returns suggests that we observe discrete, isolated liquid water pockets rather than a broad region of saturated pore space. This similarly supports our interpretation of fracture development beneath the medial trough.

**Sill Emplacement and Surface Hydrology**. We apply the TopoToolbox[73] flow accumulation analysis to the 32 m ArcticDEM surface topography product to confirm that the ridge is located within a closed basin with no route for surface water outflow in all years from 2012 to 2016. In April 2012, prior to the observed surface uplift, a lake depth of roughly 14 m at the center of the depression would have been required to overtop the basin based on ArcticDEM strip DEMs. Landsat 7 ETM+ optical imagery from the summers (June/July/August) of 2009, 2010, and 2011 show that a small surface lake formed in this basin during each melt season, likely through a combination of local melt and lateral runoff from upslope ablation (Supplementary Fig. 6 a-c). These lakes extended at most 1.3 km in length and were likely only a few meters in depth, given that in April 2012 the basin retained a typical concave bowl shape (Supplementary Fig. 2), rather than the flat bottom that would be characteristic of a refrozen ice lid over a deep buried lake.

Between April 15, 2012 and July 9, 2013, this basin experienced a surface uplift of up to 9 m (Supplementary Fig. 2 and 3). Landsat 7 imagery from July 18, 2012 shows that a flat spot had already developed within the basin and only minimal water ponding was present near the edges of this area. This stands in sharp contrast to the three other surface depressions surrounding this basin, which on this date all contained large surface lakes at the maximum extent observed in any Landsat images between 2000 and 2016. This trend continued throughout the summer of 2012, during which time no surface lake appeared to develop within the ridge basin based on available Landsat imagery (Supplementary Fig. 6d). This is particularly surprising given that 2012 was one of the most extreme melt seasons on record in Greenland[67], with record-breaking surface runoff[61] and bare ice exposure[74] in this region. Based on these observations, we infer that the majority of the surface water that flowed into this basin in the summer of 2012 drained into the subsurface through fractures in the overlying ice slabs, leading to the observed surface uplift and apparent absence of ponded surface water, while forming the sill we later observe in the 2015 radar data.

## Data availability

All processed radar sounding data used in this study are available from the CReSIS public ftp page: ftp://data.cresis.ku.edu/data/. ArcticDEM digital elevation models[48] are available from the University of Minnesota Polar Geospatial Center (PGC): https://www.pgc.umn.edu/data/arcticdem. ICESat-2 laser altimetry tracks are available through the OpenAltimetry portal at https://openaltimetry.org/data/icesat2/ with download

services provided by the National Snow and Ice Data Center. Landsat imagery is available through the USGS GLOVIS site. The Supplementary Information lists the radar, DEM, ICESat-2, and Landsat data files used in this study. The full commercial satellite image in Fig. 1 is available upon request from the Polar Geospatial Center for US researchers with funded grants from the NSF Office of Polar Programs or NASA Cryospheric Sciences through the NextView license program with the US government (https://www.pgc.umn.edu/data/commercial-imagery/). The image catalog ID is provided in the Supplementary Information. The image segment covering the double ridge has been publicly released as part of this study's data. The Galileo image of Europa shown in Fig. 1 is available on the Planetary Data System imaging node: https://pds-imaging.jpl.nasa.gov. High-resolution firn core density profiles[53,54] and conductivity measurements[56–59] used for radiometric calibration are available from PANGAEA. MARv3.5.2 climate model outputs[75] and metadata are available from the NSF Arctica Data Center at: https://arcticdata.io/catalog/view/doi%3A10.18739%2FA2H12V80V. IMAU FDM model outputs are available by request to imau@science.uu.nl due to the large data volume. Context data shown in Supplementary Fig. 1 are available as part of the QGreenland package at. The data[76] generated in this study and reprocessed radargrams required reproduce the results of this study have been deposited at Zenodo. These data consist of the WorldView image shown in Fig. 1b, the reprocessed radargrams shown in Supplementary Fig. 4 and Supplementary Fig. 5A, the reprocessed radargram underlying the phase coherence analysis detailed in Supplementary Methods Section 2, the raw inversion outputs shown in Supplementary Fig. 5C, and the raw inversion outputs underlying the sensitivity test results described in Supplementary Methods Section 5. Source data and plotting scripts to reproduce Fig. 3d and Supplementary Fig. 5c are provided in the Source Data file. Source data are provided with this paper.

## Code availability

DEM processing was conducted in the open-source software QGIS. Radargram processing was conducted using the open source CReSIS Toolbox[77]. The MATLAB scripts developed for radar analysis and inversion[78] are available at https://github.com/rtculberg/IceFractionInversion and permanently archived through Zenodo.

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

## Acknowledgements

R.C. was supported by a National Defense Science and Engineering Graduate Fellowship. R.C., D.M.S., and G.S. were supported in part by NASA Grant NNX16AJ95G and NSF Grant 1745137. We acknowledge the use of data from CReSIS and the CReSIS toolbox generated with support from the University of Kansas, NASA Operation IceBridge grant NNX16AH54G, NSF grants ACI-1443054, OPP-1739003, and IIS-1838230, Lilly Endowment Incorporated, and Indiana METACyt Initiative. Geospatial support for this work was provided by the Polar Geospatial Center under NSF-OPP awards 1043681 and 1559691. DEMs were provided by the Polar Geospatial Center under NSF-OPP awards 1043681, 1559691, and 1542736. We thank Peter Kuipers-Munneke for providing the IMAU FDM model output.

## Author contributions

R.C., D.M.S., and G.S. conceived the study. R.C. conducted the data processing and analysis. D.M.S. contributed to the development of the radar processing methods. R.C., D.M.S., and G.S. contributed to the development of the formation mechanism, the scientific interpretation of the results, and the writing of the manuscript.

## Competing interests

The authors declare no competing interests.
