## [Peer Review File · Nature Communications]

Double Ridge Formation Over Shallow Water Sills on Jupiter's Moon EuropaREVIEWER COMMENTS

Reviewer #3 (Remarks to the Author):

Dear Nature Communication editors and Dr. Culberg.

It has been a real pleasure to read this significant and well-written manuscript. The feature described here is by far the most convincing analog to European double ridges that I have seen up to now. I am happy that the Greenland ice sheet presents better analogs to Europa than the relatively freely moving ice shelves. Having been able to capture the formation of the ridge (Fig.3, S2) and characterize its sub-surface structure is remarkable. The paper presents schematically a formation mechanism that is quite reasonable and supported by the observations, although certainly more complex than I would have initially imagined. The importance of refreezing makes a lot of sense to me. I did not imagine that it would lead to the sill splitting into two domains, but I agree this explains the observations presented here well.

I refrain from commenting on the radar analysis as I am not at all a specialist on the matter. I found the manuscript generally preempted my concerns and whenever I had a question, the answer could be found a few paragraphs further. As a result, I have only a few comments that aim principally at furthering the discussion.

1) Mass conservation: The schematic diagram of Figure 4 implies that the ridge forms as the water rises from pressurized sills. It would be important to attempt a mass balance analysis. If the water movement dominates, can you document a depression or a void space left behind by the water that supports the ridge? If crystallization dominates, please estimate the volume of water that needs to crystallize to develop the ridge you observe and compare it to the sill volume. An estimate is quoted in line 111 but I couldn't find where in the methods that estimate is described. A further complication arises from the need to melt the water in the first place (especially on Europa, where there shouldn't be surface precipitation). A thaw-freeze sequence has a 0 net volume increase. How do you propose to generate new topography rather than redistributing it laterally?

2) Topography and stratigraphy: I was not familiar with the concept of refrozen ice in Greenland and I suspect others in the planetary community will not either. Could you include a few comments about the origin of that refrozen ice? On a maybe related topic, what prevents the water to drain to the base of the ice sheet (as in the case of a Moulin, which can progress unstably e.g. Krawczynski et al., 2009 10.1029/2008GL036765), which is necessary to produce a sill (Line 150: Why did the water stop draining where it did). Finally, What is the origin of the basin in which the ridge has been found. It seems too much a coincidence to have the sill and refrozen ice essentially coincident with that basin. If there is a genetic relation, it may inspire a new look at Europa data to search for evidence of this broader structure. Note the smooth terrain flanking the ridge in figure 1a towards the top of the image.

3) It has been debated whether European single ridges evolve into double ridges (Head 2000, Johnston and Montési, 2014). Here you show this progressive evolution but dismiss it. Why do you think the single ridge is originally the effect of transverse compression (L. 162). Why wouldn't it be the expression of the plug in Figure 3c?

4) Limitations of the model. I list here two aspects of Europa's geology that may be hard to explain with this model. This does not invalidate the paper and may be good motivation for further studies. However, if you have ways to address these concerns, it may be good to include them in the paper. First, the need to separate the sill into two sections (each one giving rise to a ridge) bothers me. Double ridges are so common on Europa that I find it amazing we always be able to break the reservoir into two parallel sections. Ridge length can become problematic: if the plug is not continuous along strike, water can move from one side of the plug to the other and lead to an asymmetric ridge. Second, sills in volcanic systems flatten fairly uniformly. Why would they become elongated especially with the extreme ratios implied by European double ridges?

Minor editorial comments:

- L. 45: I typically prefer spelling out "20 to 30 km" rather than using hyphens in the text. Same thing for "10-15" in line 105.

- L. 45-46: Maybe cite some of the more recent estimates of ice shell thickness, even though they are consistent with the range proposed here: Quick & Marsh (2015, 10.1016/j.icarus.2015.02.016) Peddinti & McNamara (2019 10.1016/j.icarus.2019.03.037), Green et al. (2021 10.1029/2020JE006677; Howell (2021, 10.3847/PSJ/abfe10)
 - Line 105: insert a space between numbers and units, as you do elsewhere.
 - Line 106: I am confused by the statement that the sill is “well below the depth of the overlying refrozen ice slab”. In the image, it seems to me that the sill is inside the slabs. That is confirmed by a comparison between the water table depth (10 to 15 meters) and the depth of refrozen ice quoted as 25 m on line 115. The sill is not below the refrozen ice.
 - L. 157. Missing period after “discontinuous”.
 - L. 165: Why would the edge of the plug be a plane of weakness. I might have imagined that the newly crystallize ice is quite massive, devoid of defects, and therefore strong.
 - L. 175: Replace “rich” with “promising”?
 - L. 236: Why is the regolith qualified as “tidal”. Could small impacts and/or gas release generate void spaces or fragment the ice?
 - Figure 1a: indicate the image number in the caption.
- Laurent Montesi, University of Maryland College Park.

Review Culberg et al., submitted to Nature Communications

The study by Culberg et al. focuses on a possible Greenland analogy of processes that are observed on Jupiter's moon Europa. The authors report on the discovery of a so-called double ridge on the Greenland ice sheet. Double ridges are extremely frequent on Europa where they criss-cross most of the moon's surface. If the Greenland double ridge were a suitable analogy to Europa's double ridges and were formed by similar processes, then this would provide a unique opportunity to study the phenomena.

General remarks

The conclusions of the manuscript might be valid. However, in my opinion the manuscript does provide insufficient evidence that the double ridge on Greenland is more than just accidentally similar to the ones on Europa. In particular, I miss substance in description and interpretation of processes at the Greenland site and more space needs to be dedicated to the question whether Greenland is a suitable analogy for Europa.

Greenland observations: The authors do an excellent job in processing and analysing the radar data. However, I believe that other types of data receive too little attention. The manuscript lacks some basic information. Mainly the coordinates of the observed double ridge are not provided, also Extended Data Figure 1 does not show a precise location. The exact location of the double ridge (54.0294 °W; 74.5602 °N) is within a local depression and was in four out of seven summers (2012 to 2018; Fig. 1) located below the visible surface runoff limit. Thereby we define the visible surface runoff limit as the uppermost elevation on the ice sheet where lateral runoff of meltwater is ubiquitous enough to become visible at the surface. Up to three streams or slush fields discharge into the depression which might partly turn into a lake (Fig. 1). Whether the water covers the entire depression remains unclear, the available satellite images provide no unequivocal evidence of a lake. Nevertheless, the observations mean that around, and possibly between, the double ridges, there is meltwater ponding for roughly one or even two months a year. While this observation does not necessarily invalidate the suggested process of formation of the double ridge, it needs discussion. What does the frequent presence of surface water mean in the context of the suggest process of double ridge formation? How does the frequent and prolonged presence of surface water influence validity of the Greenland-Europa analogy?

Analogy to Europa: In my opinion, the manuscript does not sufficiently discuss to what degree Greenland can serve as an analogy to Europa's surface. In particular, the manuscript makes no mentioning of certain physical properties that differ very strongly between Greenland and Europa. As far as I know, the surface temperature on Europa is around 110 K while mean annual air temperature at the site of the Greenland double ridge is roughly 260 K. The rheology of ice changes strongly between these two temperatures. Is this relevant in the context of the proposed Greenland- Europa analogy? Maybe there is an influence by the high salt concentration in the supposed subsurface ocean of Europa? Is there a potential influence by the very different atmospheric pressure on Earth and Europa? With the exception of the differences in gravity on Earth and Europa, none of these rather fundamental differences are addressed in the manuscript.

2012-07-18 Landsat 7

2014-08-03 Landsat 8

2015-08-04 Landsat 8

2016-07-30 Landsat 8

Fig. 1: Hydrological surface network at the location of the double ridge in the summers of 2012, 2014, 2015 and 2016, as seen by Landsat 7 or 8 at 30 m resolution in the near-infrared band. The red dot marks the surface depression which features the double ridge. Landsat images for the years 2012 to 2018 have been analysed for evidence of surface meltwater discharge. Shown are only the years where the hydrological network became visible at the surface. The years 2013, 2017 and 2018 did not have sufficient melt to form a visible discharge network at the site of the double ridge. The black stripes in the Landsat 7 image are due to failure of the sensor's scan-line corrector.

Detailed Comments

Lines 130 – 131: Does this statement refer to water being injected into the firn from below, that is from the sill, or to water percolating downwards from the surface?

Lines 142 – 145: At the location of the double ridge the ice sheet surface moves at about 100 m yr⁻¹. The double ridge is aligned roughly parallel to the direction of ice movement and the ICESat-2 flight track are roughly perpendicular to the double ridge. Figure 1 shows that the double ridge in Greenland varies laterally in structure and appearance. Does Figure 3d show real changes in the cross section of the double ridge or are differences simply the result of the double ridge moving by ~100 m yr⁻¹ under the ICESat-2 track?

Lines 228 – 229: Although explained in the figure caption, I suggest to provide three depth scales in Figure 3d, to avoid the impression that the surface elevation has changed substantially between 2018 and 2020.

Lines 464 – 468: The “skin depth” of firn is typically between 10 and 15 m. Below that depth, seasonal variations of firn temperature cannot be measured any more. A depth profile of firn temperatures in an ice slab region is provide in Machguth et al. (2016, supplementary material). Examples for more porous firn are provided or discussed by e.g. Benson (1959). At the location of the double ridge, the situation is complicated by prolonged seasonal ponding of meltwater at the surface. Under these conditions, the depth of zero annual temperature amplitude can be shallower.

Methods and Supplementary Material: The Methods and the Supplement provide a very thorough and clear description of the radar data processing and interpretation. I agree with the authors’ interpretation of the processed radar data. Nevertheless, the value of this thorough work is reduced by the study relying almost exclusively on the radar and the DEM data. The data need to be brought into the context of e.g. the mentioned hydrological processes. The very thorough interpretation of the radar data alone does not provide insight whether the observed phenomena is a valid analogy for Europa’s double ridges.

Supplementary figures 4 and 5: In my opinion the interpretation of these figures could be improved by providing an approximate depth scale in meters, rather than only the two-way travel time. I am aware that wave speed varies, but nevertheless would prefer some approximate indication of depth.

References

Benson, C. S. (1959): Physical Investigations on the Snow and Firn of Northwest Greenland 1952, 1953, and 1954; U.S. Army Snow Ice and Permafrost Research Establishment, Corps of Engineers.

Machguth, H., M. MacFerrin, D. van As, J.E. Box, C. Charalampidis, W. Colgan, R.S. Fausto, H.A. Meijer, E. Mosley-Thompson and R.S. van de Wal (2016): Greenland meltwater storage in firn limited by near-surface ice formation, *Nature Climate Change*, 6, 390-393.

Response to Reviewers

We would like to thank the editor for facilitating reviews from both the Europa science and Greenland surface process communities and both reviewers for their detailed and constructive comments. One of the tensions in writing this manuscript was attempting to do justice to both the terrestrial and planetary aspects of the discussion and the feedback from both areas of expertise has helped us to balance and strengthen both sides of our argument. In particular, we have worked to clarify and expand on how Greenland might serve as a useful analog for Europa, despite the many differences in environment. Our responses to the reviewers' comments are detailed below.

Reviewer #3 (Remarks to the Author)

Dear Nature Communications editor and Dr. Culberg,

[R1-1] *It has been a real pleasure to read this significant and well-written manuscript. The feature described here is by far the most convincing analog to European double ridges that I have seen up to now. I am happy that the Greenland ice sheet presents better analogs to Europa than the relatively freely moving ice shelves. Having been able to capture of formation of the ridge (Fig.3, S2) and characterize its sub-surface structure is remarkable. The paper presents schematically a formation mechanism that is quite reasonable and supported by the observations, although certainly more complex than I would have initially imagined. The importance of refreezing makes a lot of sense to me. I did not imagine that it would lead to the sill splitting into two domains, but I agree this explains the observations presented here well.*

I refrain from commenting on the radar analysis as I am not at all a specialist on the matter. I found the manuscript generally preempted my concerns and whenever I had a question, the answer could be found a few paragraphs further. As a result, I have only a few comments that aim principally at furthering the discussion.

Thank you! Your insightful comments have been very helpful in strengthening our arguments.

[R1-2] *Mass conservation: The schematic diagram of Figure 4 implies that the ridge forms as the water rises from pressurized sills. It would be important to attempt a mass balance analysis. If the water movement dominates, can you document a depression or a void space left behind by the water that supports the ridge? If crystallization dominates, please estimate the volume of water that needs to crystallize to develop the ridge you observe and compare it to the sill volume. An estimate is quoted in line 111 but I couldn't find where in the methods that estimate is described. A further complication arises from the need to melt the water in the first place (especially on Europa, where there shouldn't be surface precipitation). A thaw-freeze sequence has a 0 net volume increase. How do you propose to generate new topography rather than redistributing it laterally?*

Thank you for this suggestion – this is certainly an important sanity check on the mechanism we are proposing. We estimate the initial volume of the sill based on the observed surface displacement, which is described in the “Surface Elevation Change” portion of the Methods section at line 501. We have added a sentence at line 514 to clarify the connection between this section of the methods and the volume estimate. We think that crystallization will dominate in ridge formation, since this is the primary force that can act counter to gravity/overburden to drive water flow. Based on the 2016 DEM, the total ridge volume is approximately $1.2 \times 10^5 \text{ m}^3$. With 9% expansion by volume from freezing, approximately 18.5% +5.7% / -3.5% of the estimated sill volume would have needed to refreeze in order to produce the new ridge

volume, assuming that the expansion is not compensated in any other way. This seems completely plausible given the large refrozen ice shell surrounding the sill observed in our radar data in 2016. It is also possible that a general subsidence of the basin surface could partially compensate the ridge volume. Unfortunately, the vertical accuracy and absolute vertical registration of the available surface DEMs is not good enough to reliably assess whether this occurred over the Greenland sill, particularly given that seasonal accumulation and ablation make it extremely difficult to confidently partition meter-scale changes in surface elevation in such a small area without long-duration, detailed field measurements. To clarify these points for the reader, we have added much of this discussion to the manuscript at lines 204-212.

The question of mass conservation in sill formation on Europa is an interesting one. Assuming that the general scale relationships we observe in Greenland hold for Europa, the sills required to support double ridges are significantly larger than the ridges themselves. Therefore, it is possible that over length scales similar to the sill's horizontal extent (10s to 100s of kms depending on the dimension) there would be observable surface uplift (in the case of sill injection from the subsurface ocean, assuming that it is injected into a horizontal fracture rather than filling pore space) or subsidence (in the case of internal melt pockets). Certainly, in Greenland we saw up to 9 m of local uplift within the basin where the sill was emplaced. It is possible that these kinds of large-scale, more subtle variations in topography might also be detectable for some of the youngest ridges on Europa, although the many generations of cross-cutting ridges at the surface would likely complicate the interpretation. In general, we have assumed in this manuscript that sill emplacement on Europa is reasonable and do not significantly interrogate exactly how that would occur or its implications for surface deformation given that Greenland is a poor analog for this component of the process and other numerical and analytical studies have considered it in much more detail already. We have added a section on discussing the analogy between Europa and Greenland in more detail at line 213, which may help clarify this point for the reader.

[R1-3] *Topography and stratigraphy: I was not familiar with the concept of refrozen ice in Greenland and I suspect others in the planetary community will not either. Could you include a few comments about the origin of that refrozen ice?*

Thanks for pointing this out. We have added some clarifying descriptions of the processes by which subsurface refrozen ice might form in Greenland. Specifically, lines 101-110 now describe the typical subsurface density structure in this region and the origin of the near-surface ice slabs that form through percolation and refreezing of surface meltwater.

[R1-4] *On a maybe related topic, what prevents the water to drain to the base of the ice sheet (as in the case of a Moulin, which can progress unstably e.g. Krawczynski et al., 2009 10.1029/2008GL036765), which is necessary to produce a sill (Line 150: Why did the water stop draining where it did).*

From a terrestrial perspective, this is actually a fascinating question to study and we believe quite important for understanding the water budget in NW Greenland. Glaciologists have largely studied moulins that form in the ice sheet ablation zone, where the entire ice column consists of solid, low-permeability ice. When a surface crevasse fills with water, the water pressure can compensate the lithostatic stresses that act to close the crevasse, thereby allowing it to penetrate deeper – in some cases all the way to the bed of the ice sheet (Van Der Veen, 1998). The region of the ice sheet where this double ridge is located has a rather different near-surface structure that consists of 3 to 5 m of solid refrozen ice known as an ice slab at the surface, overlying a layer of porous, compacting firn that eventually transitions to solid ice again at a depth of 30 to 40 m below the surface (M. MacFerrin et al., 2019; Machguth et al.,

2016). What this suggests is that once a water-filled crevasse penetrates the ice slab, water can now leak out of the crevasse into the surrounding porous firn, reducing the water pressure, and therefore stresses, at the fracture tip and potentially arresting propagation. At this location, the resistive tensile stresses that might open surface fractures are not particularly large to begin with (likely less than 200 kPa as estimated from surface velocity-derived strain rates) and so we suspect that this reduction in water pressure is likely sufficient to reduce the combined tensile and water-pressure induced stress to be less than the lithostatic stresses driving closure, although we would need to develop a coupled model of flow and fracture mechanics to definitively confirm this.

The other possibility is that a complete surface-to-bed connection is formed, but the rate of surface meltwater input is so extreme relative to the transmissivity of the subglacial environment that water essentially backs up within the moulin or fracture and then begins to spread laterally from the fracture walls into the surrounding porous firn. While we do not have sufficient observational evidence to rule out this possibility, we think it is a somewhat less likely mechanism, given that meltwater lakes routinely drain to the bed in the ablation zone over the course of hours (Das et al., 2008) and other work has suggested that low transmissivity at the bed is compensated by the formation of a water 'blister' between the ice and bedrock (Lai et al., 2021; Stevens et al., 2015) rather than a backing up of the conduit.

Regardless, the general process of water storage in porous firn beneath ice slabs seems to be incredibly common in this region of Greenland. We have separately documented more than 100 instances of subsurface refrozen ice masses in radar sounding data from this region, and associated many of them with high elevation moulins or lake drainage in the 2012 extreme melt season. However, we felt that a detailed discussion of this process would be outside the scope of this manuscript and detract from our focus on the Europa analogy, given that the method by which sills form in Greenland and Europa will be dramatically different and the analog is most useful for understanding what happens next (i.e., the impact of recrystallizing sills on surface morphology). That said, we have added a brief sentence describing the leak-off mechanism at line 170 to preempt this question to some extent.

[R1-5] *Finally, what is the origin of the basin in which the ridge has been found. It seems too much a coincidence to have the sill and refrozen ice essentially coincident with that basin. If there is a genetic relation, it may inspire a new look at Europa data to search for evidence of this broader structure. Note the smooth terrain flanking the ridge in figure 1a towards the top of the image.*

The surface topography in this region of Greenland is quite varied and consists of a many small hills and basins that reflect the underlying bedrock topography, filtered through the viscous deformation of about 1500 m of overlying ice. Given the stable location of this basin over time (i.e., it is not advected over the surface with ice flow), it is most likely an expression of bed topography. The presence of this basin was almost certainly critical to the initial formation of the sill, because meltwater would flow laterally over the surface into this hydraulic low, allowing it collect sufficient water to initiate hydrofracture and transport water into the subsurface. However, given that the drivers of surface topography and sill formation mechanisms are very different on Europa, we think the basin is a component of the system that is unlikely to translate to the European setting. However, we have amended our description of the Greenland sill formation process at line 165 to acknowledge the role of this basin in surface meltwater collection.

[R1-6] *It has been debated whether European single ridges evolve into double ridges (Head 2000, Johnston and Montési, 2014). Here you show this progressive evolution but dismiss it. Why do you think the single ridge is originally the effect of transverse compression (L. 162). Why wouldn't it be the expression of the plug in Figure 3c?*

Thank you for this comment. We think that it is difficult to disentangle the mechanisms that might be involved in the transition from single to double ridge, especially given our lack of detailed information on ice flow in this basin. We do agree it is possible that the single to double ridge transition might be a feature of this subsurface water configuration and refreezing. For example, higher initial pressure in one side of the split sill might lead to the asymmetric system we initially observe with a tall northern ridge and very small southern ridge. We think it is unlikely that the single ridge would be a direct expression of the ice plug (i.e, surface extrusion from the fracture or uplift over the refreezing plug) because it is difficult to explain how this 2 m tall ridge would later subside to form the medial trough between the double ridges which are themselves only ~2 m in height. Overall, while we do observe an evolution from single to double ridge, it is not clear that the mechanism we propose must first form a single ridge in order for a double ridge to develop. As a result, we did not want to overstate the potential connection but seem to have inadvertently overemphasized transverse compression as a result. We have amended this part of the discussion at lines 186-193 to acknowledge that our observations suggest that single ridges can be an early stage of double ridge formation, but that the mechanism we propose does not imply that single ridge formation must occur, and that it is difficult to disentangle the mechanisms that might have led to this evolution in the case of the Greenland ridge.

[R1-7] *Limitations of the model. I list here two aspects of Europa's geology that may be hard to explain with this model. This does not invalidate the paper and may be good motivation for further studies. However, if you have ways to address these concerns, it may be good to include them in the paper. First, the need to separate the sill into two sections (each one giving rise to a ridge) bothers me. Double ridges are so common on Europa that I find it amazing we always be able to break the reservoir into two parallel sections. Ridge length can become problematic: if the plug is not continuous along strike, water can move from one side of the plug to the other and lead to an asymmetric ridge.*

We actually think that partial hydrologic connection between the two halves of the sill is not implausible and might actually help guarantee the observed ridge symmetry. Under our proposed mechanism, symmetric ridge development should require equivalent overpressure conditions within the sill on either side of the central refrozen conduit, since this is the force that drives mass displacement and extrusion at the surface. One way to maintain that equilibrium would be for the two side of the divided sill to remain partially hydrologically connected – for example, if the refrozen ice plug did not extend the full depth of the sill (see, for example, how we represent the plug in Fig 4c and d) or if, as you suggest, the plug is not continuous along-strike or does not extend the full length of the sill. We think this might be one reason why the Greenland system was able to develop a relatively symmetric ridge, despite the fact that the two sides of the divided sill area have clearly different volumes and extents (see Fig. 2). We now explicitly discuss this possibility at line 182-186.

Consistently breaking sills into two parallel sections comes back to the question that seems to have plagued many hypotheses for double ridge formation – is it possible to routinely form continuous linear fractures 100s of km in long on Europa? Our work here does not bring any new insights on that question. However, we do note in the discussion that this is a still poorly understood requirement of our mechanism and that in order to this to be true, Europa's ice shell would need remarkable lateral consistency in its rheology and composition (see line 313).

[R1-9] *Second, sills in volcanic systems flatten fairly uniformly. Why would they become elongated especially with the extreme ratios implied by European double ridges?*

We think there are three possibilities that might be consistent with our mechanism. First, a ridge might form over only part of a roughly circular (in plan view), but very large sill. Our observations in Greenland seem to suggest that ice plug does not need to perfectly bisect the sill to still achieve ridge symmetry – the observed double ridge actually initially forms on the northern edge of the sill (see Supplementary Figure 2) and after division, the southern sill segment is still about twice the length of the northern segment (see Figure 2). This scenario might be able to form ridges on the order of a few 10s of kms long, given that chaos terrain has previously been associated with subsurface water bodies up to 80km in diameter. However, this is perhaps less likely, given that the very large water bodies that may support chaos terrain appear to induce complete surface collapse in some cases.

Second, the sill formation mechanism might be such that uniform flattening does not produce a circular shape. For example, suppose the sill is formed when subsurface ocean water rises through the ice column inside a fracture. If the horizontal extent of the fracture is small relative to its width, such that it can be treated like a point source, then we would expect a typical penny-shaped sill. However, if the fracture extent is extremely long relative to its opening width (for example, 100 km long fracture with a few meters opening width), then we might expect even spreading of the fluid to form more of an elongated pill shape that reflects the geometry of the initial fracture.

Third, sills might be the expression of local melt over top of rising linear diapirs. In this case, the length to width ratio of the sill would be roughly governed by that of the warm diapir driving the melt.

We have expanded and clarified our discussion of these three possible mechanisms at lines 300-311 in the discussion.

Minor editorial comments:

[R1-10] L. 45: I typically prefer spelling out “20 to 30 km” rather than using hyphens in the text. Same thing for “10-15” in line 105.

Amended in text.

[R1-11] L. 45-46: Maybe cite some of the more recent estimates of ice shell thickness, even though they are consistent with the range proposed here: Quick & Marsh (2015, 10.1016/j.icarus.2015.02.016) Peddinti & McNamara (2019 10.1016/j.icarus.2019.03.037), Green et al. (2021 10.1029/2020JE006677; Howell (2021, 10.3847/PSJ/abfe10)

Thank you for these suggestions. We have included these citations in the revised text.

[R1-12] *Line 105: insert a space between numbers and units, as you do elsewhere.*

Amended in text.

[R1-13] *Line 106: I am confused by the statement that the sill is “well below the depth of the overlying refrozen ice slab”. In the image, it seems to me that the sill is inside the slabs. That is confirmed by a comparison between the water table depth (10 to 15 meters) and the depth of refrozen ice quoted as 25 m on line 115. The sill is not below the refrozen ice.*

Thanks for pointing this out – I believe this confusion comes from our use of the term “ice slab” to refer to a very specific type of subsurface refrozen ice complex that forms in Greenland. Specifically, “ice slabs” have been defined as approximately 1 to 15 m thick horizontal layers of refrozen ice that form just beneath the annual snow accumulation layer when surface meltwater percolates into the subsurface and refreezes (MacFerrin et al., 2019). They are understood to be horizontally continuous over tens of kilometers and remained perched above otherwise porous firn. So while the sill we observe is indeed surrounded by a shell of refrozen ice, the morphology and likely formation mechanism of that refrozen mass do not fit with the traditional definition of “ice slabs”. The point we wanted to make in this sentence is that the subsurface water pocket we observe must have originally formed within the porous firn layer because the depth at which we observe water is significantly deeper than the typical thickness of perched ice slabs in this region (usually 3 to 5 m). We have added a clarifying paragraph at lines 100-110 that explains the typical subsurface structure in this region of the ice sheet (ice slab overlying porous firn) and the origin of these features.

[R1-14] *L. 157. Missing period after “discontinuous”.*

Amended in text.

[R1-15] *L. 165: Why would the edge of the plug be a plane of weakness. I might have imagined that the newly crystallized ice is quite massive, devoid of defects, and therefore strong.*

We think that the edges of the plug would be planes of weakness because they likely represent a rheologic discontinuity or transition between the ice plug, which as you note would likely be strong and devoid of defects, and the surrounding refrozen ice shell which was formed by the slow refreezing of water that infiltrated the complicated porous firn matrix. We agree that the plug itself would be unlikely to fracture, so if stresses are still concentrated in the general area of the plug, fracturing might be shifted to the weaker ice to its left or right. We have added a brief note about this rheologic transition at line 197.

[R1-16] *L. 175: Replace “rich” with “promising”?*

Amended in text – this discussion now occurs in the “Greenland as a Europa Analog” section at line 215.

[R1-17] *L. 236: Why is the regolith qualified as “tidal”. Could small impacts and/or gas release generate void spaces or fragment the ice?*

We had initially considered tidal regolith because – once scaled to Europa – our mechanism probably requires appreciable porosity in the upper few kilometers of the ice shell. Small impacts or gas release seems like less likely mechanisms at those depths (relative to the upper few meters). However, your point

is well taken that mechanisms controlling ice shell porosity are very poorly constrained and we have removed the “tidal” qualifier in the text.

[R1-18] *Figure 1a: indicate the image number in the caption.*

Amended in text.

Review Culberg et al., submitted to Nature Communications

The study by Culberg et al. focuses on a possible Greenland analogy of processes that are observed on Jupiter’s moon Europa. The authors report on the discovery of a so-called double ridge on the Greenland ice sheet. Double ridges are extremely frequent on Europa where they criss-cross most of the moon’s surface. If the Greenland double ridge were a suitable analogy to Europa’s double ridges and were formed by similar processes, then this would provide a unique opportunity to study the phenomena.

General remarks

[R2-1] *The conclusions of the manuscript might be valid. However, in my opinion the manuscript does provide insufficient evidence that the double ridge on Greenland is more than just accidentally similar to the ones on Europa. In particular, I miss substance in description and interpretation of processes at the Greenland site and more space needs to be dedicated to the question whether Greenland is a suitable analogy for Europa.*

Thank you for these comments. We had some difficulty in deciding how deeply we should discuss some of the Greenland processes given the planetary focus of the paper and your review has been a significant help in guiding how and where we fleshed out those details. Specifically, we have made the following changes which are discussed more extensively in response to your detailed comments below:

- 1) We have expanded our discussion of the role of surface hydrology in forming the initial sill in Greenland (lines 164, 244-270, and 606-630).
- 2) We have added an additional section to the manuscript at line 213 titled “Greenland as a Europa Analog” where we discuss how environmental differences between Earth and Europa, including gravity, atmospheric pressure, temperature, ice chemistry, and the presence or absence of surface processes like melt, impact the proposed ridge formation mechanism.

Please see our responses to the detailed comment below for more in-depth discussion of each of these changes.

Greenland observations:

[R2-2] *The authors do an excellent job in processing and analysing the radar data. However, I believe that other types of data receive too little attention.*

Thank you for suggesting this additional analysis. As part of our expanded discussion on the role of surface hydrology in this system, we now include an analysis of the available Landsat imagery over our time period of interest and include some of the key images in a new Supplementary Figure 6. Please see our detailed responses below for an extended discussion of these additions.

[R2-3] *The manuscript lacks some basic information. Mainly the coordinates of the observed double ridge are not provided, also Extended Data Figure 1 does not show a precise location.*

Thanks for catching this omission. We have added the center coordinates of the double ridge in the main text at line 89 and have updated Extended Data Figure 1 to include an overlay of visible imagery from Sentinel-2, the outline of the double ridge, and graticules on the inset showing latitude and longitude. We also include the latitude and longitude in the figure caption.

[R2-4] *The exact location of the double ridge (54.0294 °W; 74.5602 °N) is within a local depression and was in four out of seven summers (2012 to 2018; Fig. 1) located below the visible surface runoff limit. Thereby we define the visible surface runoff limit as the uppermost elevation on the ice sheet where lateral runoff of meltwater is ubiquitous enough to become visible at the surface. Up to three streams or slush fields discharge into the depression which might partly turn into a lake (Fig. 1). Whether the water covers the entire depression remains unclear, the available satellite images provide no unequivocal evidence of a lake. Nevertheless, the observations mean that around, and possibly between, the double ridges, there is meltwater ponding for roughly one or even two months a year. While this observation does not necessarily invalidate the suggested process of formation of the double ridge, it needs discussion. What does the frequent presence of surface water mean in the context of the suggest process of double ridge formation? How does the frequent and prolonged presence of surface water influence validity of the Greenland-Europa analogy?*

Thank you for highlighting this gap. In the process of writing the original manuscript, we had spent some time analyzing the available Landsat imagery at the double ridge site and discussing the hydrologic context, but ultimately did not include the details in the final manuscript in an effort to keep it succinct and focused on Europa. However, it is clear that in doing so we ended up leaving out an important piece of context and discussion that was necessary to support the analogy we wished to draw. We have addressed this issue in the revised manuscript in the following ways:

- 1) We have added Supplementary Figure 6 which includes a time series of Landsat 7 and 8 images from the summers of 2009 through 2016 that highlight the surface hydrology in each year at the double ridge site and surrounding area, bracketing the years with available radar data. Additionally, in this figure we also include a plot of integrated annual runoff from this site as modeled by MARv3.10 for additional context. This data makes it clear, as you described, that local surface ponding is frequent during the melt season, but the extent can vary significantly from year to year.
- 2) When describing the initial emplacement of the sill at line 164, we now highlight that one of the key lines of support for our hypothesis that the sill was formed by surface runoff draining through fractures in the overlying ice slab is the Landsat imagery record from 2012. The 2012 summer imagery shows little to no ponding within the double ridge basin, despite the record high melt and the fact that all of the surrounding basins developed deep surface lakes that were larger than observed in any other year between 2000 and 2016. Given that ponding had occurred in the double ridge basin in 2009-2011, we believe this is further evidence (besides the surface uplift)

that most runoff in summer 2012 was routed to the subsurface within this basin. We have added an additional section in the Methods at line 604 in order to fully discuss the imagery observations and our interpretations while keeping the main text succinct.

- 3) We have added a section that addresses how environmental differences between Greenland and Europa might impact our analogy. This includes discussion of the role that ponded or flowing surface water might play in shaping the observed ridge morphology and how it might interact with the subsurface processes we infer from the radar data (lines 244-270). We come to the following conclusions which we discuss in the main text:
 - a) Surface water might be able to recharge the sill on a seasonal basis if old surface fractures are reactivated or new fractures develop and then heal over the winter. This would not necessarily alter the mechanism that we propose, but it might reduce the amount of total refreezing required to form the added volume of the ridges, as additional external water volume would be added to the sill.
 - b) Surface water flow or ponding could lead to spatially variable ablation rates that could alter the surface topography in the basin. Based on the available DEMS, the ridges are located in a closed basin with very low surface slopes and relatively little variation in surface topography (besides the ridges). This suggests that stream incision is unlikely to be a significant process, since stream velocity, slope, and depth would all be small. Enhanced ablation beneath ponded or slow flowing water due to the lower albedo could be significant. However, since this is a closed basin, we expect that such a process would largely redistribute mass laterally, since water that melts within the basin will almost certainly refreeze somewhere else within the basin. The fact that water tends to pond between or next to the ridges suggests there might be additional ablation in those locations, but since the ponding location is consistent across most Landsat images, it likely refreezes there as well for net zero mass exchange. Regardless, there does not seem to be any easy way that water flow alone could uniformly downcut an area of nearly 3 km² on the basin floor, leaving only the isolated ridges that stand 2 meters above basin floor. However, water flow could play a role in forming the troughs we observe to either side of the ridges in the 2017 radar data.
 - c) In general, accumulation and ablation will likely alter the morphology of the ridges to some extent. However, we argue that this would likely result in less ridge symmetry and more variation in ridge structure along-strike, suggesting that the absence of these processes on Europa would favor the formation of the very consistent ridges we observe there and might explain the greater variations seen in the Greenland double ridge. In general, we cannot think of a combination of surface accumulation and ablation processes alone that would produce the ridge morphology observed, absent any other factors. We also argue that the relatively consistent location, length, and cross-sectional form of the ridge from 2016 through 2020 despite large variations in annual accumulation, melt, and runoff from year to year suggests that these processes play an ancillary role in shaping ridge morphology.

Altogether, we think that the frequent and prolonged presence of surface water makes Greenland a poor analog for the emplacement or formation of sills on Europa. Therefore, we emphasize to the reader that this part of the process should not be considered as analogous and that Greenland is most useful for providing examples of the physical behavior associated with the refreezing of isolated water pockets within a porous near-surface layer, similar to the configurations hypothesized for Europa. We also suggest

that interaction between surface and subsurface water would not fundamentally change the mechanism we propose, and that there is not clear mechanism for producing the ridges from surface processes alone, though surface processes likely have some difficult to quantify impact the precise morphology and dimensions of the system.

Analogy to Europa:

[R2-5] *In my opinion, the manuscript does not sufficiently discuss to what degree Greenland can serve as an analogy to Europa's surface. In particular, the manuscript makes no mentioning of certain physical properties that differ very strongly between Greenland and Europa. As far as I know, the surface temperature on Europa is around 110 K while mean annual air temperature at the site of the Greenland double ridge is roughly 260 K. The rheology of ice changes strongly between these two temperatures. Is this relevant in the context of the proposed Greenland- Europa analogy? Maybe there is an influence by the high salt concentration in the supposed subsurface ocean of Europa? Is there a potential influence by the very different atmospheric pressure on Earth and Europa? With the exception of the differences in gravity on Earth and Europa, none of these rather fundamental differences are addressed in the manuscript.*

Thank you for highlighting the importance of discussing these differences between Earth and Europa. We have added a new section in the manuscript at line 213 that discusses how environmental differences between the two planets might impact Greenland's usefulness as a Europa analog. We think it is important to note that we do not start this paper from the assumption that conditions on Greenland and Europa are analogous and therefore any similar features we observe should have the same underlying physical mechanism. Rather, we discovered a double ridge in Greenland with similarities to double ridges on Europa and after digging into the data, found some conditions similar to some of those hypothesized to exist at Europa – namely, a subsurface water body emplaced in porous ice and slowly refreezing. Therefore, one of the (perhaps surprising) implications of our results is that Greenland has the potential to be useful for observing the fundamental physical processes that occur when confined, isolated water bodies refreeze within a partially porous ice matrix.

Of course, how the behaviors of such systems scale between these different environments may be complicated, as you describe. Therefore, in this added section, we discuss how differences in atmospheric pressure, gravity, ice rheology (due to differences in temperature and ice impurity content), and surface processes (e.g. melt) might impact the specific mechanism we propose for double ridge formation. It is important to keep in mind that some of these variables are very poorly constrained for Europa – particularly temperature and ice impurity content – due to the lack of observations. Quantitatively characterizing the impact of Europa's ice rheology on ridge formation would require a whole ensemble of complicated thermomechanical numerical model runs for many different plausible scenarios, which is far beyond the scope of this manuscript (although it would be a fascinating follow-on study). Instead, we qualitatively discuss whether these factors are likely to fundamentally alter or prevent the ridge formation mechanism we propose, or whether they would simply modulate the length and time scales over which ridge formation could occur.

In summary, we come to the following conclusions:

- 1) Reduced gravity and atmospheric pressure (relative to Earth) should aid ridge formation by reducing overburden and therefore relaxing the overpressure condition needed to drive water upwards within the fractures.

- 2) What little we can constrain about the subsurface structure of Europa suggests that sills of the type we propose would be firmly located with the brittle, conductive portion of the ice shell where either elastic deformation or fracture would dominate the response to imposed stress. This is consistent with our mechanism, which invokes fracture and the extrusion or uplift of damaged material, and does not require viscous creep. Numerous thermomechanical modeling efforts have also shown that, in a general sense, surface disruption from the crystallization of shallow water bodies is possible under a range of temperature and pressure conditions that would be reasonable for Europa.
- 3) Surface processes like melt and water flow cannot easily explain the formation of the ridges without some other subsurface processes at play (see detailed response to R2-4), and heterogeneous ablation should reduce the symmetry and along-strike consistency of the Greenland ridge, making it less, rather than more, like Europa's ridges. Therefore, we suggest that the subsurface processes we observe are the primary drivers of the surface ridge formation, rather than surface processes which would not be possible on Europa.

Detailed Comments

[R2-6] Lines 130 – 131: *Does this statement refer to water being injected into the firn from below, that is from the sill, or to water percolating downwards from the surface?*

This was intended to refer to water from the sill being forced upwards into the overlying firn due to overpressure in the sill. We have clarified this point in the text at line 141.

[R2-7] Lines 142 – 145: *At the location of the double ridge the ice sheet surface moves at about 100 m yr⁻¹. The double ridge is aligned roughly parallel to the direction of ice movement and the ICESat-2 flight track are roughly perpendicular to the double ridge. Figure 1 shows that the double ridge in Greenland varies laterally in structure and appearance. Does Figure 3d show real changes in the cross section of the double ridge or are differences simply the result of the double ridge moving by ~100 m yr⁻¹ under the ICESat-2 track?*

Thank you for pointing this out. We agree that it is not possible to tell from our data whether the cross-section of the double ridge (as shown in Fig. 3d) has changed with time or whether the observed changes are simply a result of a laterally varying structure that is advected through the repeated flight lines over time. We now discuss this explicitly at line 156, as well as noting this point in the caption for Figure 3.

[R2-8] Lines 228 – 229: *Although explained in the figure caption, I suggest to provide three depth scales in Figure 3d, to avoid the impression that the surface elevation has changed substantially between 2018 and 2020.*

Thanks for this suggestion, we have updated the figure accordingly.

[R2-9] Lines 464 – 468: *The “skin depth” of firn is typically between 10 and 15 m. Below that depth, seasonal variations of firn temperature cannot be measured any more. A depth profile of firn temperatures in an ice slab region is provide in Machguth et al. (2016, supplementary material). Examples for more*

porous firn are provided or discussed by e.g. Benson (1959). At the location of the double ridge, the situation is complicated by prolonged seasonal ponding of meltwater at the surface. Under these conditions, the depth of zero annual temperature amplitude can be shallower.

Thank you for suggesting these references. Our phrasing in the original text was perhaps unclear – we were attempting to convey that the observed scatterers are located a sufficiently shallow depth that they would be subject to seasonal temperature variations and, if representative of water pockets, we would therefore expect them to at least partially refreeze seasonally. It seems that this is in general agreement with temperature profiles in Machguth (2016) which we interpret as showing the coldest seasonal temperatures occurring around 1 m depth with significant latent heating from refreezing in the fall and early winter at depth of around 5 meters. However, you make a good point that the thermal situation at this location is complicated – both the seasonal surface ponding and the slow refreezing of a very large subsurface water body mean that the subsurface temperature profile might deviate significantly from the typical profiles in literature. We have edited this section of the methods (lines 585-593) to clarify the points discussed above.

[R2-10] Methods and Supplementary Material: *The Methods and the Supplement provide a very thorough and clear description of the radar data processing and interpretation. I agree with the authors' interpretation of the processed radar data. Nevertheless, the value of this thorough work is reduced by the study relying almost exclusively on the radar and the DEM data. The data need to be brought into the context of e.g. the mentioned hydrological processes. The very thorough interpretation of the radar data alone does not provide insight whether the observed phenomena is a valid analogy for Europa's double ridges.*

As discussed in detail in response to R2-4, we have added analysis of the available Landsat imagery in order to provide additional context for the surface hydrology at our study site. This includes a more thorough description of the role of surface hydrology in forming the sill (lines 164 and 604) and a discussion of the potential impact of standing water on ridge formation and evolution (lines 244-270).

[R2-11] Supplementary figures 4 and 5: *In my opinion the interpretation of these figures could be improved by providing an approximate depth scale in meters, rather than only the two-way travel time. I am aware that wave speed varies, but nevertheless would prefer some approximate indication of depth.*

Thanks for this suggestion. In Supplementary Figure 4, elevation compensation distorts the hyperbolic scattering that we wish to emphasize, so we have instead added an approximate depth scale in meters (assuming a wave velocity in solid ice throughout the subsurface) as an additional y-axis on the figure, with the 0 m point set as the mean surface elevation along the profile. For Supplementary Figure 5, we have done full elevation correction and replaced the two-way travel time axis with an absolute elevation axis, since the large variations in surface elevation make a single depth-below-surface axis impractical.

References

Benson, C. S. (1959): Physical Investigations on the Snow and Firn of Northwest Greenland 1952, 1953, and 1954; U.S. Army Snow Ice and Permafrost Research Establishment, Corps of Engineers.

Machguth, H., M. MacFerrin, D. van As, J.E. Box, C. Charalampidis, W. Colgan, R.S. Fausto, H.A. Meijer, E. Mosley-Thompson and R.S. van de Wal (2016): Greenland meltwater storage in firn limited by near-surface ice formation, *Nature Climate Change*, 6, 390-393.

Response References

Das, S. B., Joughin, I., Behn, M. D., Howat, I. M., King, M. A., Lizarralde, D., & Bhatia, M. P. (2008). Fracture Propagation to the Base of the Greenland Ice Sheet During Supraglacial Lake Drainage. *Science*, 320(May), 778–782.

Lai, C.-Y., Stevens, L. A., Chase, D. L., Creyts, T. T., Behn, M. D., & Stone, H. A. (2021). Hydraulic transmissivity inferred from ice-sheet relaxation following Greenland supraglacial lake drainages. *In Revision, Nature Communications*, 1–10. <https://doi.org/10.1038/s41467-021-24186-6>

MacFerrin, M. J., Machguth, H., van As, D., Charalampidis, C., Stevens, C. M., Heilig, A., ... Abdalati, W. (2019). Rapid expansion of Greenland's low-permeability ice slabs. *Nature*, 573, 403–407. <https://doi.org/10.1038/s41586-019-1550-3>

Machguth, H., MacFerrin, M., Van As, D., Box, J. E., Charalampidis, C., Colgan, W., ... Van De Wal, R. S. W. (2016). Greenland meltwater storage in firn limited by near-surface ice formation. *Nature Climate Change*, 6(4), 390–393. <https://doi.org/10.1038/nclimate2899>

Stevens, L. A., Behn, M. D., McGuire, J. J., Das, S. B., Joughin, I., Herring, T., ... King, M. A. (2015). Greenland supraglacial lake drainages triggered by hydrologically induced basal slip. *Nature*, 522(7554), 73–76. <https://doi.org/10.1038/nature14480>

Van Der Veen, C. J. (1998). Fracture mechanics approach to penetration of surface crevasses on glaciers. *Cold Regions Science and Technology*, 27(1), 31–47. [https://doi.org/10.1016/S0165-232X\(97\)00022-0](https://doi.org/10.1016/S0165-232X(97)00022-0)

REVIEWER COMMENTS

Reviewer #3 (Remarks to the Author):

I want to thank the authors for a clear and complete answer to my comments. The expanded discussion of the analogy between Greenland and Europa is particularly helpful. I have only minor (typographic) comments, which focus on the updated text.

Line 89: Missing degree symbols on coordinates

Line 157 and 398: Please check the using and format in “106 ma-1”

Line 206: Missing “to” in “would need to refreeze TO form”

Line 207: No dash “Widespread”

Line 236: missing “the” in “within THE brittle, conducting portion”

Line 237: The citation number 8 could be mistaken with a power on the unit K. I think there is a different style for this case, maybe “150 K (8)”

Line 302: Missing “of”. “the development OF both sills...”

I would also strongly encourage the authors to make official release of their codes and obtain a permanent citation using the process described at <https://docs.github.com/en/repositories/archiving-a-github-repository/referencing-and-citing-content>

Laurent Montesi

Reviewer #4 (Remarks to the Author):

I would like to thank the authors for addressing my review of the first version of this manuscript. I have no more comments on the study, which I believe is suitable for publication.

I just wonder about one point, which could be highlighted a bit more: In the manuscript, lines 173 and following, it is stated that initial cracks are generated by the sill freezing from its upper boundaries inward. However, looking at Supplementary Figures 2 and 3, I have the impression that the initial crack roughly marks the outer margin of the area of strong uplift (by up to 9 m from 2012 to 2013). Hence, I imagine the initial crack as a mechanical failure of the firn, possibly also including the ice slab. In this interpretation, the uplift due to water inflow caused the crack, refreezing processes might initially not have been important. Afterwards, the crack was advected towards the west by the ice flow and found itself close to the middle of the basin in 2016. The interesting thing is indeed how it has transformed into a double ridge and then preserved its structure during that movement.

Response to Reviewers

Reviewer #3 (Remarks to the Author):

[R3-1] *I want to thank the authors for a clear and complete answer to my comments. The expanded discussion of the analogy between Greenland and Europa is particularly helpful. I have only minor (typographic) comments, which focus on the updated text.*

Thank you again for our thoughtful comments that significantly improved this manuscript.

[R3-2] *Line 89: Missing degree symbols on coordinates*

Fixed in text.

[R3-3] *Line 157 and 398: Please check the using and format in “106 ma-1”*

Here we have followed the style guidance from the International System of Units that ‘year’ should be abbreviated as ‘a’ (annum) when used with SI units. But we are happy to defer to any specific style guidance from the editors on this abbreviation.

[R3-4] *Line 206: Missing “to” in “would need to refreeze TO form”*

Fixed in text.

[R3-5] *Line 207: No dash “Widespread”*

Fixed in text.

[R3-6] *Line 236: missing “the” in “within THE brittle, conducting portion”*

Fixed in text.

[R3-7] *Line 237: The citation number 8 could be mistken with a power on the unit K. I think there is a different style for this case, maybe “150 K (8)”*

Amended in text to “150 K (ref. 8)”

[R3-8] *Line 302: Missing “of”. “the development OF both sills...”*

Fixed in text.

[R3-9] *I would also strongly encourage the authors to make official release of their codes and obtain a permanent citation using the process described at <https://docs.github.com/en/repositories/archiving-a-github-repository/referencing-and-citing-content>.*

Thanks for this reminder! We waited to archive the code in case any revisions were requested as part of the review process, but we have now done so and the final version is now archived under doi: 10.5281/zenodo.5865616.

Reviewer #4 (Remarks to the Author):

[R4-1] *I would like to thank the authors for addressing my review of the first version of this manuscript. I have no more comments on the study, which I believe is suitable for publication.*

Thank you again for all of your detailed comments – they helped us greatly in strengthening our arguments and balancing the terrestrial and planetary perspectives in this manuscript.

[R4-2] *I just wonder about one point, which could be highlighted a bit more: In the manuscript, lines 173 and following, it is stated that initial cracks are generated by the sill freezing from its upper boundaries inward. However, looking at Supplementary Figures 2 and 3, I have the impression that the initial crack*

roughly marks the outer margin of the area of strong uplift (by up to 9 m from 2012 to 2013). Hence, I imagine the initial crack as a mechanical failure of the firn, possibly also including the ice slab. In this interpretation, the uplift due to water inflow caused the crack, refreezing processes might initially not have been important. Afterwards, the crack was advected towards the west by the ice flow and found itself close to the middle of the basin in 2016. The interesting thing is indeed how it has transformed into a double ridge and then preserved its structure during that movement.

Thanks for this comment! Indeed, it seems very possible that the initial fracture we observe in 2013 was the result of mechanical failure during injection and uplift. However, based on our radar observations, the splitting of the sill by refreezing did not occur until sometime after the spring of 2015. This suggests that this fracture may have initially healed within this compressional basin and later been reactivated, perhaps due to overpressure within the sill. We have now clarified these possibilities and our uncertainty in the precise mechanism and timeline of fracture formation in the text at lines 166 and 171 (line number refer to the main text version with tracked changes).